# Exploring tumor-normal cross-talk with TranNet: Role of the environment in tumor progression

**Bayarbaatar Amgalan[1], Chi-Ping Day[2], Teresa M. Przytycka[1] ***

**1** National Center for Biotechnology Information/National Library of Medicine, National Institutes of Health, Bethesda, Maryland, United States of America, **2** Laboratory of Cancer Biology and Genetics/Center for Cancer Research/National Cancer Institute, National Institutes of Health, Bethesda, Maryland, United States of America

* przytyck@ncbi.nlm.nih.gov

**Data Availability Statement:** The authors confirm that all data underlying the findings are fully available without restriction. There are no primary

## Abstract

There is a growing awareness that tumor-adjacent normal tissues used as control samples in cancer studies do not represent fully healthy tissues. Instead, they are intermediates between healthy tissues and tumors. The factors that contribute to the deviation of such control samples from healthy state include exposure to the tumor-promoting factors, tumor-related immune response, and other aspects of tumor microenvironment. Characterizing the relation between gene expression of tumor-adjacent control samples and tumors is fundamental for understanding roles of microenvironment in tumor initiation and progression, as well as for identification of diagnostic and prognostic biomarkers for cancers.

To address the demand, we developed and validated TranNet, a computational approach that utilizes gene expression in matched control and tumor samples to study the relation between their gene expression profiles. TranNet infers a sparse weighted bipartite graph from gene expression profiles of matched control samples to tumors. The results allow us to identify predictors (potential regulators) of this transition. To our knowledge, TranNet is the first computational method to infer such dependencies.

We applied TranNet to the data of several cancer types and their matched control samples from The Cancer Genome Atlas (TCGA). Many predictors identified by TranNet are genes associated with regulation by the tumor microenvironment as they are enriched in G-protein coupled receptor signaling, cell-to-cell communication, immune processes, and cell adhesion. Correspondingly, targets of inferred predictors are enriched in pathways related to tissue remodelling (including the epithelial-mesenchymal Transition (EMT)), immune response, and cell proliferation. This implies that the predictors are markers and potential stromal facilitators of tumor progression. Our results provide new insights into the relationships between tumor adjacent control sample, tumor and the tumor environment. Moreover, the set of predictors identified by TranNet will provide a valuable resource for future investigations.

data in the paper; all software is available at https://github.com/ncbi/TranNet.

**Funding:** This research was supported by the Intramural Research Program of the National Library of Medicine (T.M.P and B.A), and Intramural Research Program of the Center for Cancer Research, National Cancer Institute (C.-P. D). The funders had no role in study design, data collection and analysis, decision to publish, or preparation of the manuscript.

**Competing interests:** The authors have declared that no competing interests.

## Author summary

In oncological studies, control samples are usually biopsied from tumor-adjacent normal tissue. However, there is an increasing understanding that such samples represent a state that is intermediate between tumor and normal, and is influenced by environmental factors common to tumor and normal tissues, and by tumor microenvironment. Therefore, uncovering the relation between gene expressions across control and tumors samples can inform us about the roles of microenvironment in tumor initiation and progression. Here we present a predictive model, TranNet, to study the functional relationship between matched control and tumor samples. TranNet infers a transition function from gene expression in a control sample to that in the matched tumor sample. Simultaneously, the method identifies a set of genes that are predictors of this transition. To our knowledge, TranNet is the first computational method to infer such dependencies.

Our results demonstrated that TranNet efficiently captured the relation between tumors and their microenvironment, generating important implications for the detection, diagnosis, and prognosis of cancers.

## Introduction

In multi-stage carcinogenesis theory, mutations accumulate to perturb the cell regulatory program, eventually causing cell transformation. These perturbations interact with other factors such as micro and macro environment or preexisting health conditions. Although mutations are central for the emergence of cancer, much of the understanding of the disease comes from studies of the cancer-related changes in gene expression. Cancer is characterized by dysregulated functions of many cellular processes including proliferation, cell-cell interactions, chromatin organization, DNA repair, and others. Yet many of these alterations are not mechanically linked to specific mutations, but are driven by changes in gene expression [1]. One of the emerging concepts is that cancer progression is facilitated by increased cell plasticity, which allows cancer cells to switch dynamically between a differentiated states in response to stress [2]. Cancer cell plasticity has been linked to the epithelial-to-mesenchymal transition (EMT) which has been shown to respond to microenvironmental signals or cancer therapy [3–8]. In addition, it has been estimated that at least 25% of cancers are associated with chronic inflammation [9, 10]. Last but not least, a recent study suggested that the tumor microenvironment and the microenvironment of control sample are strongly dependent [11]. In fact, tumor-related alteration of adjacent tissue are believed to contribute to postoperative cancer recurrences that occur in up to a third of patients [12]. This is not a surprise, as tumors and their adjacent normal tissues also share some exogenous exposures including environmental factors, such as smoking, diet, and genetic variations. However could gene expression from a control sample inform on tumor state? Which molecular pathways and functions in tumor are associated with gene expression changes in normal tissue? What can they teach us about tumor progression?

To address these questions, we developed the Transition Network model (TranNet), a computational approach to study the relation between the gene expression patterns in matched normal and tumor tissues. Focusing on tumor genes (defined as genes differently expressed between tumor and control samples), TranNet infers a transition function from gene expression in a control sample to an estimate of gene expression in the matched tumor sample. Simultaneously, the method infers a set of genes that are predictors of this transition (Fig 1).

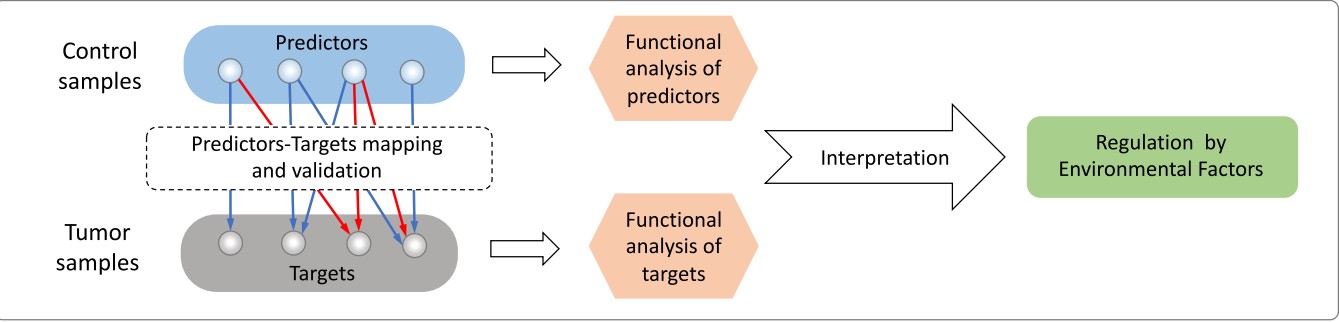

**Fig 1. Workflow of the analysis.** TranNet constructs a transition function from gene expression in control samples to gene expression in the matched tumor samples. Simultaneously, the method infers a set of predictors in the control tissue that used for computing the transition and target genes in the tumor tissue that are influenced by the predictors. After validating the method, functional analysis of the predictors and their target genes is used to shed light on possible sources of the relation between gene expression in control and tumor samples.

These predictor genes might but do not have to regulate this transition, and might instead reflect expression changes due to factors common to control and tumor samples. Functional analysis of the predictors and their target genes helps to shed light on possible sources of this relation.

Computationally, TranNet utilizes a network construction based on a sparse estimation of partial correlation [13] that uses an $l_1$ norm constraint to ensure the selection of the most informative predictors (Fig 2). Recognising that the expression of the genes which did not pass the p-value threshold to be included in the set of tumor genes might also contribute to gene expression in tumor, we extended the network by including additional nodes, principal components, representing the informative trends on the expression data of these genes. We refer to the genes and principal components whose activity in control samples influence gene expression in tumor tissue samples as *predictors*.

TranNet opens a new way to explore the relationship between gene expression in adjacent control samples and tumors. Our results indicate that the former can provide information about the latter and link, at least in part, inferred relations to tumor environment. As elaborated in Section 'Discussion and conclusions', not all inferred associations are assumed to be tumor drivers. Yet, as predictors of the expression of tumor genes, they are markers of other tumor-related processes including tumor-environment interaction and are important in this respect. Indeed, using TranNet we identified a set of genes that can serve as the predictors of tumor-environment relation as well as genes and pathways involved in this interaction. Taken together, this work offers a computational method to infer the relation between the gene expression patterns in matched normal and tumor tissues, and provided a new understanding of the relation between tumor, normal, and the environment.

## The transition network model (TranNet)

An overview of the TranNet method is presented in Fig 2. Two matrices representing gene expressions in normal (A1) and tumor (A2) tissues of cancer patients are provided as the input. Tumor genes, defined as genes differentially expressed between control and tumor samples, are represented as network nodes while the expression of the rest of the genes is represented by meta-nodes corresponding to the principal components of the expression of these remaining genes. Each gene or principal component is represented by two nodes—one for each condition. The conceptual idea of including the principal components in the analysis is illustrated in (B). The transition matrix from normal to tumor state is obtained by solving the

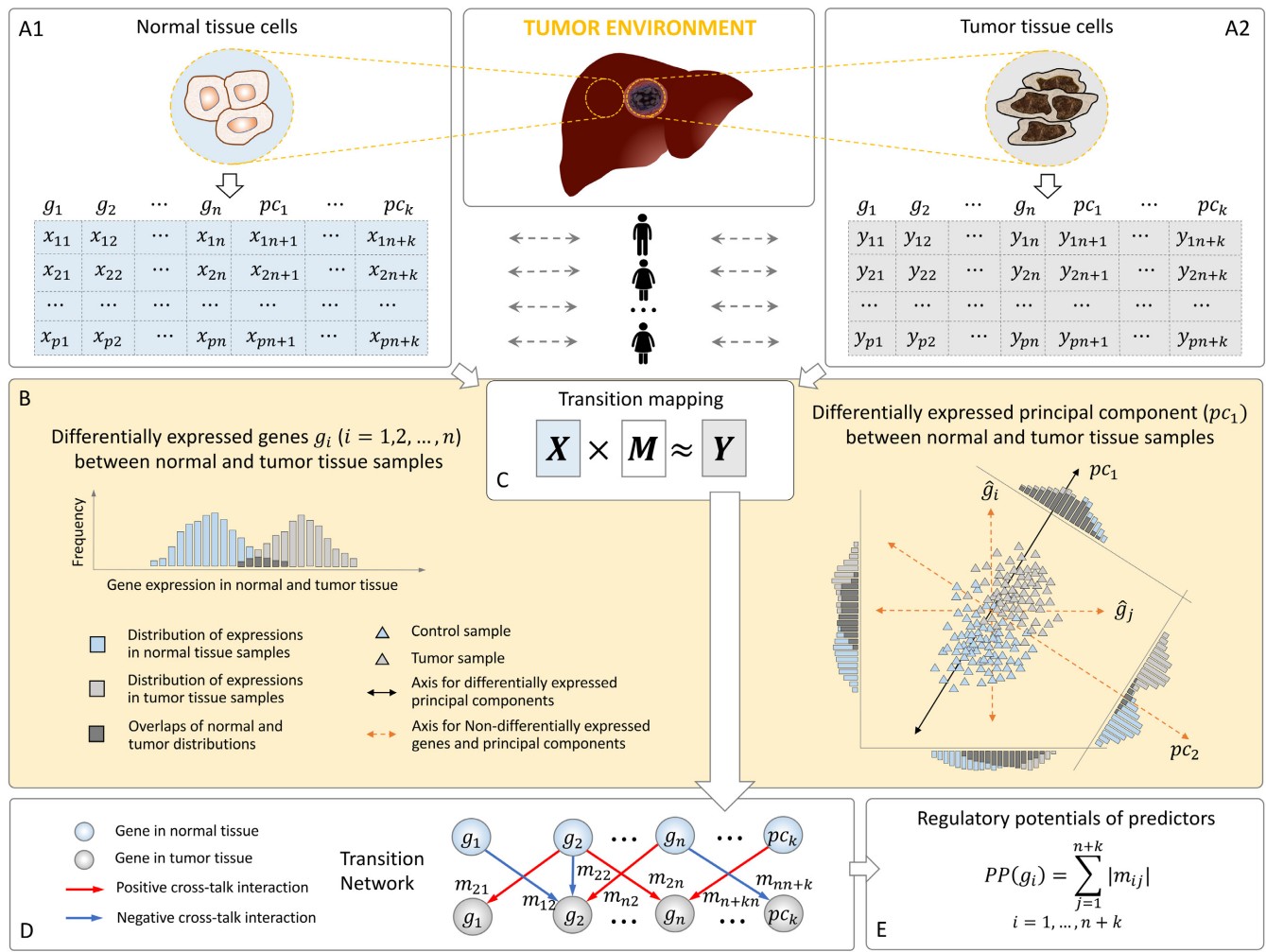

**Fig 2. Outline of the TranNet method.** Input matrices $X$ and $Y$ represent the expressions of genes in control ($X$, panel (**A1**)) and tumor ($Y$, panel (**A2**)) samples across $p$ patients (rows). Genes ($g_i$, $i = 1, \ldots n$) which are differentially expressed between control and tumor samples (referred to as tumor genes) are represented by individual nodes (columns from 1 to $n$) and the rest of the genes are represented by principal components of their expression (columns from $pc_1$ to $pc_k$) representing the major trends of the expression. The principal components which differentiate between normal and tumor tissue samples are considered as additional variables in the multi-variate analysis. The distribution trend of the expression of a differentially expressed gene $g_i$ is visualized in the left of the yellow panel. As illustrated in a 2-dimensional space in the right of the yellow panel, although neither $\hat{g}_i$ nor $\hat{g}_t$ is differentially expressed between control and tumor tissue samples, their joint distribution is differentiated along the axis of principal component $pc_1$ showing a similar trend to the DE genes while $pc_2$ is not differentiated (**B**). The transition map is a linear operator defined by matrix $M$ computed to minimise $Y$'s representation error subject to a sparsity constraint explained in the main text (**C**). The result is summarised as a bipartite network representing explanatory influences from control samples to tumors (**D**). Predictive potential of a gene represents its total contribution to the transition network (**E**).

problem in (C) and the bipartite network formed by the transition matrix represents information flow from normal to tumor tissues (D). Finally, the strength of the total influence of a node in normal tissue on gene expression in tumor is quantified by predictive potentials (PP) (E).

## Inference of the transition network

Let $X$ and $Y$ be the matrices describing gene expression in tumor and control samples respectively for $p$ patients. The first $n$ columns of each matrix represent the expression of the $n$

tumor genes followed by $k$ meta nodes corresponding to the principal components summarising gene expression trends of the remaining set of genes.

Specifically, $x_{ij}$ and $y_{ij}$ denotes the expression of gene $g_j$ (or $pc_{j-n}$ if $j > n$) in the $i^{th}$ patient in tumor and normal respectively where the "expression" of a principal component in patient $i$ is explained below (see also Fig 2B right, for the description of differentially expressed (DE) principal component).

Assuming that $X$ represents regulatory influences (or markers of such influences) and $Y$ their targets, the weight matrix $M$ describing transformations between them can be written as

$$X \times M \approx Y. \tag{1}$$

More precisely, for patient $i$, the expression value $y_{if}$ of gene $g_f$ or principal component $pc_f$ in the patient's tumor tissue can be approximated by a linear combination of the expressions of the $n$ genes and $k$ meta nodes in the patient's normal tissue

$$\sum_{j=1}^{n+k} x_{ij} \cdot m_{jf} \approx y_{if}$$

where $m_{(\cdot)f} \in \mathbb{R}^{n+k}$ denotes the transition weights from the normal tissue expression of the $n$ genes and $k$ principal components to the tumor tissue expression of gene $g_f$ or principal component $pc_f$. Thus, our goal is to minimize the least square error subject to a unit $l_1$ norm constraint on $m_{(\cdot)f}$ as follows:

$$\underset{m_{(\cdot)f} \in R^{n+k}}{\text{minimize}} \quad \sum_{i=1}^{p} \left( \sum_{j=1}^{n+k} x_{ij} \cdot m_{jf} - y_{if} \right)^2, \quad \text{subject to} \quad \sum_{j=1}^{n+k} |m_{jf}| \leq 1. \tag{2}$$

The $l_1$ norm constraint on the edge weights allows for selecting the strongest transition effects on a given target node. Hence, this regularization acts to avoid over-fitting issues and only non-zero $m_{jf}$, selected for $g_f$, denotes the transition effect from $g_j$. In this setting, for every target node, the optimization problem in Eq (2) (a constrained version of Eq (1)) searches over all possible combinations of transition effects from the predictor nodes and selects the best combination with their optimal weights to explain the activity of the target node [13].

To explain the role of principal components we shall recall that a fundamental assumption for multivariate analysis is that there are no unobserved factors affecting both explanatory and response variables globally. The expressions of the genes that were not selected as tumor genes (DE genes) might have such influence. To adjust for such effects on the transition mapping, we include meta nodes (adjustment variables in [14]) that represent the principal components of the expression data for these non-tumor genes (see Section 'Materials and methods'). By the definition, principal components are vector representations of the general trends in a multi-dimensional data [15]. Here data points represent all the samples (control and tumor) and each point is a vector representing gene expression (Fig 2B right). Embedding this data using principal components as reference axis, we say that a given principal component is differentially expressed between the control and tumor samples if the coordinates of the tumor samples on the axis defined by this principal component are significantly larger or smaller than the coordinates of the control samples (Fig 2B right).

Finally, we comment on the optimization problem. Although the least square minimization in Eq (2) is convex, the $l_1$ norm constraint is non-smooth and derivative-based optimality conditions such as Lagrangian multipliers and Karush–Kuhn–Tucker (KKT) conditions are not, in general, directly applicable. While a coordinate descent algorithm with convex penalties [16] is commonly used to approximate this non-smooth problem, there is no optimal strategy

for tuning parameters controlling the strength of the penalty term [17]. Similarly complementing the standard quadratic programming formulation, the soft shrinkage method decomposes the $l_1$ norm into $2^{n+k}$ inequality constraints [18]. However, handling $2^{n+k}$ constraints is not practical for the large-scale problem. To overcome this issue, we implemented the projected gradient method [19] that converges to the optimal solution for the non-smooth constrained problem.

## Predictive potentials

TranNet aims to identify predictors (genes and principal comments) whose expression in one condition (here normal) predicts gene expression in a different condition (here tumor). We model the information flow from normal to tumor tissue by the transition matrix in Eq (1), obtained by solving the non-smooth convex optimization problem in Eq (2). The total contribution of an individual gene or a principal component to the transition is captured by the non-zero transition weights from the corresponding node to its selected targets in the network, and the total incoming effect on each target is normalized by the $l_1$ norm constraint in Eq (2). We define the *predictive potential* (PP) of such predictor as the summation of the absolute weights of the outgoing edges from the node representing the corresponding gene/principal component in normal tissue.

$$PP(g_j) = \sum_{i=1}^{n+k} |m_{ji}|. \tag{3}$$

Thus, the predictors can be prioritized with respect to their predictive potential defined in Eq (3).

## Validation of the TranNet model

We applied the TranNet model to the TCGA gene expression data for five different cancer types, focusing on solid tumors selected based on availability of matched control-tumor samples: Breast Cancer (BRCA), Lung Adenocarcinoma (LUAD), Lung squamous cell carcinoma (LUSC), Prostate Adenocarcinoma (PRAD), and Liver Hepatocellular Carcinoma (LIHC) (see Section 'Materials and methods' for data processing). To validate TranNet model, we first show that the impact of gene expression in normal tissue on gene expression in tumor is higher than expected by chance (predictability). Next, we show that the network edges inferred by TranNet are enriched in functional interactions. Subsequently, we analyse gene-wise prediction accuracy. These validations in BRCA are collected in Fig 3, and extended figures including the validation results for all the five cancers are provided in Figs A and B and C and D in S1 Text. Finally, we confirm that the results are stable, thus the conclusions of this study do not depend on the threshold used for selecting the gene set for analyzing.

### Expression of genes in control samples informs gene expression in tumor

The motivation for the TranNet model is the hypothesis that gene expression in control samples carry relevant information on gene expression in matched tumors. In order to confirm that this is indeed the case, we used a leave-one-out strategy (see Section 'Materials and methods') to test if the model can predict the expression of tumor genes from the expression of predictors better than expected by chance. The paired-sample T-test is used to compare the results (prediction errors) of two predictions: the first based on the real expression data and the second based on permuted data. In addition, we tested how prediction accuracy depends on the number of used predictors assuming that the markers are selected in the decreasing

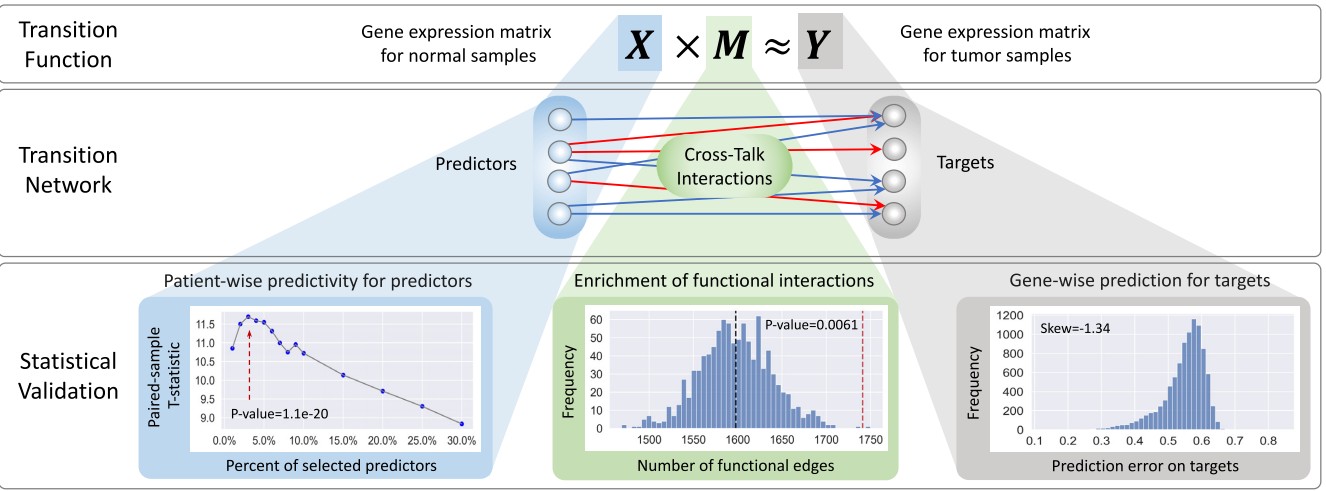

**Fig 3. Schematic representation of the analysis and validation.** Sample-wise predictivity of TranNet was evaluated based on a leave-one-out test (blue-shaded, Section 'Expression of genes in control samples informs gene expression in tumor'); Interactions inferred by TranNet were tested with enrichment of the functional interactions (green-shaded, Section 'The transition edges inferred by TranNet are enriched in functional interaction edges'); Gene-wise prediction accuracy of TranNet was evaluated by the prediction error (mean absolute error) on the targets. (grey-shaded, Section 'Expression of genes in control samples informs gene expression in tumor').

order of their PP values. The prediction accuracy varies with the selection of different percentages of the top PP scoring predictors (see Patient-wise predictability in Fig 3 and Fig A in S1 Text, and Table 1). We note that, similar to the other biological networks, the distribution of both degrees and predictive potentials in the TranNet networks is characterised by a large number of predictors with very small predictive potentials and small proportion of predictors with high potentials (see Fig E in S1 Text and S1 File). In particular, the sum of predictive potentials for the top predictors (Table 1) were 42.2% (BRCA), 46.6% (LUAD), 50.7% (LUSC), 56.5% (PRAD) and 29.5% (LIHC) of the total predictive potentials over all predictors in the respective cancers.

As expected, not all tumor genes are well predicted. However, for the top 20% of predicted genes, the coefficients of determination are above 0.6 and the Pearson correlation coefficients are close to 0.9 in all cancers expect BRCA (Fig B in S1 Text). The relatively worse performance of the method for BRCA is likely to be related to a smaller impact of environmental factors on breast cancer relative to the environmental impact on lung, liver and prostate cancers. Importantly, the error distributions are highly non-symmetric and negatively skewed towards small

**Table 1. Leave-one-out validation for the TranNet Model.** The paired-sample T-test was used to compare the sample-wise prediction errors on real and permuted response data. This test was performed for the different numbers of top PP scoring predictors. The selected predictors corresponding to the optimal cut-off are provided with their PP scores in S1 File. The bottom rows report result using top 30% of predictors. The top rows report the results for the cancer-specific optimised set of predictors (See Fig 3 and Fig A in S1 Text).

| Cancer | BRCA | LUAD | LUSC | PRAD | LIHC |
|---|---|---|---|---|---|
| Number/Percentage of predictors (optimized set) | 279/3% | 347/4% | 398/4% | 300/4% | 127/2% |
| Paired-sample T-statistic (optimized set) | 11.7 | 11.1 | 8.3 | 8.7 | 9.0 |
| Paired-sample P-value (optimized set) | 1.1e-20 | 8.7e-16 | 5.6e-11 | 2.6e-11 | 4.5e-11 |
| The number of predictors (top 30%) | 2792 | 2607 | 2989 | 1919 | 2254 |
| Paired-sample T-statistic (top 30%) | 8.8 | 7.5 | 6.1 | 6.0 | 6.5 |
| Paired-sample P-value (top 30%) | 2.6e-14 | 4.9e-10 | 1.6e-07 | 2.7e-07 | 8.1e-08 |

errors as visualized in Fig 3 and Fig C in S1 Text, suggesting that the expression of only some tumor genes can be predicted from the expression in control samples. The GO terms enriched for the gene lists sorted by the prediction errors on the genes (tumor tissue) are summarized in S2 File and discussed in Section 'Pathway enrichment of genes identified as targets of predictors revealed tumor-stroma interactions'.

## The transition edges inferred by TranNet are enriched in functional interaction edges

TranNet infers a sparse bipartite graph modelling the transition from the expression of predictors to the expression of tumor genes in matched tumor samples. Since these edges encode potential regulatory influences (or markers associated with such influences), we expect that they should be enriched for functional interactions. To test this, we computed the overlap of the TranNet edges with the edges in the human functional interaction network [20] (see Section 'Materials and methods') and tested whether this overlap is larger than expected by chance. To this end, we constructed 1,000 random networks by permuting target nodes of the inferred transition network without changing the topology of the network. The enrichment of the functional interactions [20] for BRCA is provided in Fig 3, and the performances for all the five cancers are summarized as Fig D in S1 Text. These results confirm that TranNet infers relations consistent with the functional interactions [20].

## Stability of TranNet

Since the networks inferred by TranNet are sparse (see the table in Fig E in S1 Text) and inferred from expression data using some cut-offs (see Section 'Materials and methods'), it is important to test if an alternative selection of tumor genes would not lead to very different results. To test this, we computed the transition matrix for an alternative definition of tumor (DE) and Non-DE genes where using a q-value cut-off of 0.001 in the T-test for differential expression. Then, we compared the optimal set of predictors for the main set with the same number of top predictors computed for the alternative set. The Jaccard similarity and hyper-geometric test results between the two selected sets of predictors showed very high agreements as summarized in Table 2.

## Insights into the relation between gene expression in tumor and in matched control samples

After validating TranNet model, we investigated the properties of the inferred interactions between control and tumor tissue samples. As noted above, the gene expressions are not predicted equally well for all genes. However, the error distributions are highly non-symmetric

**Table 2. Stability of TranNet.** Overlap in the selected predictors based on the networks constructed form two different selection of tumor genes. Main set is described in Section 'Materials and methods' and the Alternative set was selected with a different threshold as described in Section 'Stability of TranNet'. We selected the same number of top predictors from each list based on the number of optimal predictors for the Main set. The overlap reports the number of common predictors, Jaccard Index: Jaccard Index similarity between the two sets of the highest scoring predictors and HG $p$-value is hyper-geometric p value for the enrichment of the predictors obtained with the Alternative set in the Main set.

| | Main set | Alternative set | Selected-predictors | Overlap | Jaccard Index | HG $p$-value |
|---|---|---|---|---|---|---|
| BRCA | 9309 | 8172 | 279 | 228 | 0.8 | $\approx 0.0$ |
| LUAD | 8691 | 7225 | 347 | 282 | 0.805 | $\approx 0.0$ |
| LUSC | 9965 | 8635 | 398 | 317 | 0.79 | $\approx 0.0$ |
| PRAD | 6399 | 4464 | 127 | 73 | 0.57 | 8.7e-125 |
| LIHC | 7514 | 5806 | 300 | 215 | 0.711 | 3.5e-322 |

and negatively skewed towards small errors as visualized in Fig 3 for BRCA (the error distributions for all the five cancers are provided as Fig C in S1 Text). We started by investigating which biological pathways are enriched within the genes with better predictions. Towards this end we performed enrichment analysis for the gene list ranked by prediction accuracy. Strikingly, the genes whose expression was predicted with higher accuracy were enriched with specific groups of GO categories related to: environment, cell communication, immune response, signalling, and cell cycle (see the complete list of enriched GO terms in S2 File). Particularly, the enriched pathways in BRCA included cell-to-cell signalling, ion transport, and regulation of hormone level. Pathways in LUAD and LUSC had many terms related to cilia as expected due to the known impact of smoking on the properties of ciliated cells [21]. Pathways in LUAD were generally related to cell cycle. Finally pathways enriched in LIHC included, in addition to the pathways related to cell cycle, pathways related to detoxification and taxis and thus related to liver-specific relation with environment [22].

Interestingly, the third principal components PC3 in LUAD and PRAD, are significantly differentiated between control and tumor samples with q-values 2.10E-05 for LUAD and 4.72E-05 for PRAD (see Table A in S1 Text) and have relatively high predictive potentials ranked 104-th in the LUAD predictors and 37-th in the PRAD predictors (see S1 File). This demonstrates that expression of genes that did not pass the significance level of differential expression have effects on the differentially expressed genes. As for the targets genes of these components, both are enriched with terms related to metabolism and DNA repair, and Spliceosome (see Table A in S1 Text) while also containing cancer type specific two terms.

## Predictor genes are enriched in pathways that imply the features of tumor microenvironment

To find the properties of the inferred predictor genes, we used GOrilla [23] to perform a GO enrichment analysis. For each cancer, we identified predictor genes, ranked them by their predictive potential (PP) scores, and performed a ranked list enrichment analysis [23]. We considered all three levels: enrichment of biological process (P), molecular function (F) and cellular component (C) (See Fig 4).

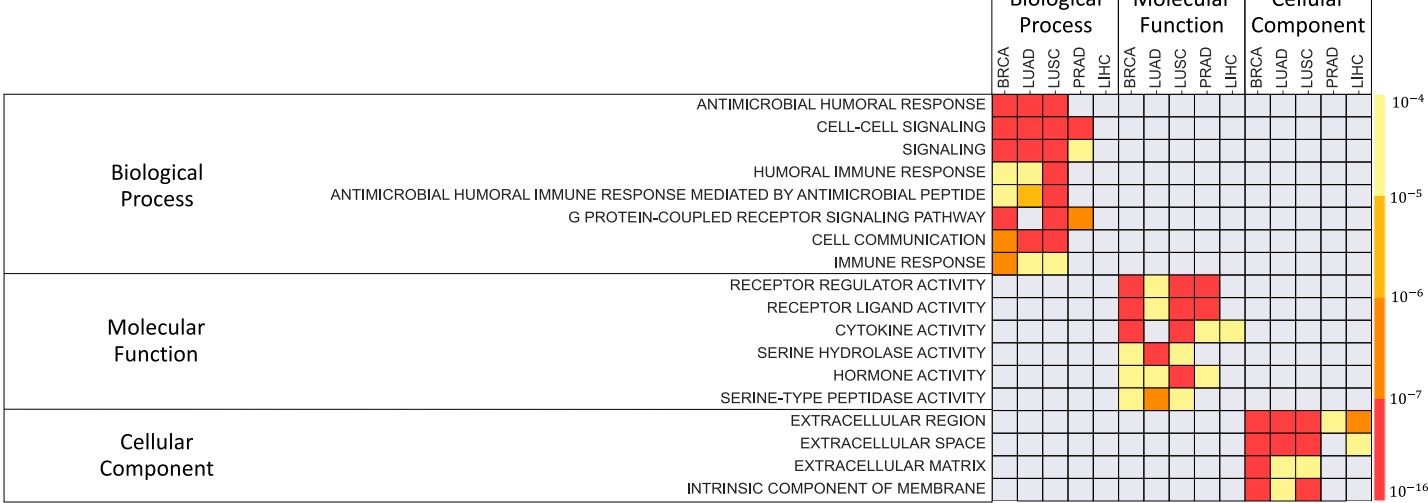

**Fig 4. Enrichment of GO terms for the predictor genes.** Rows correspond to enriched terms, while the color represents enrichment p-value. GO terms enriched in at least three cancers are shown. The complete lists of GO enrichment for the lists of predictor genes sorted by their PP scores are provided as in S3 File.

**Table 3. Top 10 PP scoring genes.** The lists of the top 10 genes with the highest predictive potentials (Eq (3)) in each cancer together with references to sample papers discussing their roles in cancer. Stars denote references to papers that point to relations with the same cancer type. The predictive potentials of the top predictors for the five cancers are provided in S1 File.

| BRCA | | LUAD | | LUSC | | PRAD | | LIHC | |
|---|---|---|---|---|---|---|---|---|---|
| NNAT | [48]* | LY6K | [49]* | ENTPD6 | [38]* | LRRC23 | [50] | ATP2A1 | [51] |
| DPEP1 | [52]* | OLAH | [53] | RASGEF1A | [54]* | PRMT5 | [45]* | TMEM253 | |
| CLEC4G | [55] | LYPD1 | [56] | SLCO1C1 | [57] | NTRK2 | [58]* | GSTA4 | [59]* |
| CCDC102B | [24]* | SLC13A4 | [60] | OR2W3 | [61] | ADAD2 | [62] | FLVCR1 | [63]* |
| GLYATL1 | [64]* | CD244 | [65]* | SH2D5 | [66] | FAM86B1 | | IGHMBP2 | [67] |
| DLEU1 | [68]* | TSACC | | B3GAT1 | [69] | TMEM178A | | CENPI | [70] |
| ADRA2B | [30]* | TMEM52 | [71]* | SLC51B | [72] | ROPN1B | [73] | CHEK2 | [74]* |
| SPOCD1 | [75] | WDR62 | [76]* | CITED1 | [77]* | LRRC7 | [50] | RNF125 | [78]* |
| BATF2 | [79]* | ELMO1 | | GNLY | [80]* | LMX1B | [81]* | AUNIP | [82] |
| EPYC | [83]* | PCDH7 | [84]* | GAL | | C2orf73 | | OAZ3 | [85] |

The predictor genes for all the five cancer types are enriched in pathways that imply the feature of the microenvironment around the tumors. For the Biological Process, they include humoral immune response, cell-cell signaling and cell communication, and G-protein coupled receptor. Molecular functions were enriched with cell communication (receptor regulator activity and receptor ligand activity), hormone activity, and inflammation (cytokine activity, serine hydrolase activity, serine-type peptidase activity). With respect to the terms related to cellular component, all cancers were enriched in pathways related to extracellular space and membrane. When compared among individual cancer types, the predictor genes have shown distinct features of tumor microenvironment. This is illustrated by the top 10 predictor genes in each cancer type (Table 3, the top 10 cut-off was selected based on 50% dropout in predictive potential). Unprecedented 85% of these genes have been identified in literature as cancer related and generally associated with tumor progression. For BRCA, the predictors include inflammation-induced genes (CCDC102B [24], SPOCD1 [25], EPYC [26], GLYATL1, SLC6A9 [27]), immune cell markers (CLEC4G [28] and BATF4 [29]), and neurogenesis genes (ADRA2B [30] and NNAT [31]). For LUAD, in addition to inflammation-induced genes (LYPD1 [32]), immune cell markers (CD244 [33]), and neural genes (WDR62 [34]), the predictors also include invasiveness-related genes (ELMO1 [35], LY6K [36], TMEM52 [37], TSACC, PCDH7). Similarly, the predictors of LUSC include to inflammation-induced genes (ENTPD6 [38], SLC51B [39], GNLY, SLCO1C1 [40], and B3GAT1 [41]), invasiveness-related genes (CITED1, OR2W3 [42], RASGEF1A [43]), and neural genes (GAL and SH2D5). Importantly, most of these genes are involved in chronic obstructive pulmonary disease (COPD). Taken together, these results suggested that BRCA, LUAD, and LUSC exhibited inflammatory neuroepithelial reactive stroma that are regulated by different mechanisms in immune responses and tissue regeneration. Also shown in Fig 4, the predictors of PRAD and LIHC were enriched in distinct pathways from other cancer types. Their features are demonstrated by the function of the top-ranked predictor genes. For PRAD, the majority of the top-ten predictors are neurogenesis genes (NTRK2 [44], LRRC7, LMX1B, ROPN1B); the others include genes involved in inflammatory response (TMEM178A) and androgen receptor regulation (PRMT5 [45]). This is relevant to the intense neuroepithelial interactions in prostate cancer. For LIHC, the predictors include genes responsive to DNA damage (AUNIP, CHEK2, CENPI) and hypoxia (FLVCR1), and two genes involved in liver disease such as fatty liver and hepatitis

(GSTA4, ATP2A1 [46], RNF125 [47]). They reflect the metabolic modification in the tumor microenvironments.

## Pathway enrichment of genes identified as targets of predictors revealed tumor-stroma interactions

To investigate how the microenvironments might regulate the tumors, we identified the pathways enriched (see Section 'Materials and methods' for the pathway enrichment) by the tumor genes ("targets"; Matrix Y in Fig 2B) associated with the top-ranked predictor genes in Table 3.

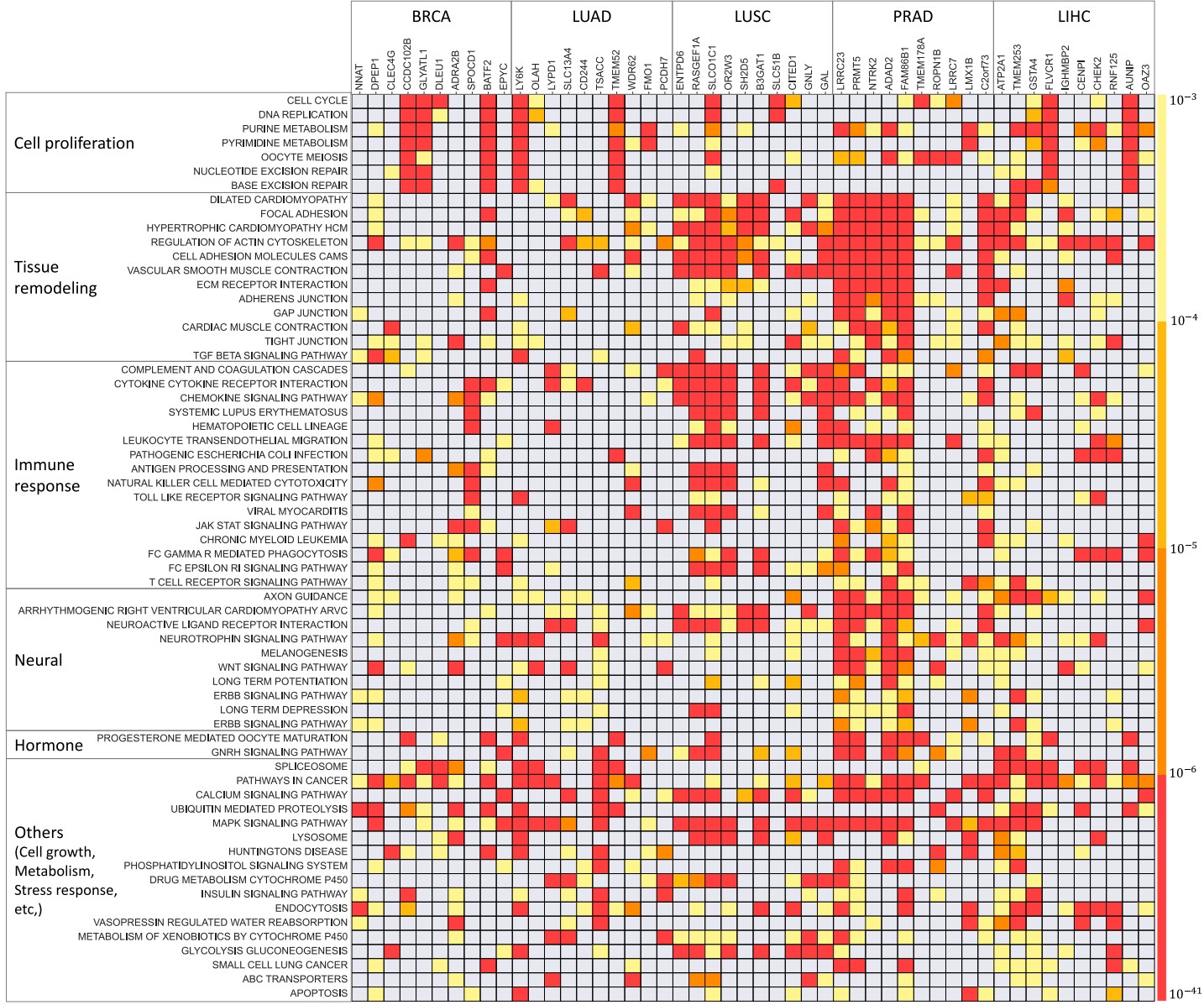

**Fig 5. KEGG pathways for target genes.** Kegg pathways enriched for the targets of the top 10 regulators (Table 3) in each cancer type. Rows correspond to the predictor genes while columns represent the pathways enriched for more than 10 predictor genes over all the five cancers. The complete lists are combined in S4 File.

As shown in Fig 5, the targets in BRCA and LUAD are more enriched in cell proliferation-related pathways (the first group in Fig 5). In LUSC, the majority of targets are enriched in tissue remodeling and immune response pathways (the second and third groups in Fig 5, respectively). In PRAD, in addition to immune response and tissue remodeling, the targets are mostly enriched in neural and hormone pathways (fourth and fifth groups in Fig 5, respectively). In LIHC, the targets are enriched mostly in cell proliferation and tissue remodeling pathways. These results reflect the distinct signaling for tumor initiation and expansion in each cancer type. For example, studies have shown that the occurrence of LUSC and LIHC are frequently preceded with Chronic Obstructive Pulmonary Disease (COPD) and hepatitis, respectively, and the tissue injury and remodeling in such chronic inflammation promote tumorigenesis. Interestingly, the same types of pathways in tumors could be associated with different types of predictors, depending on cancer type. For example, cell proliferation-related pathways in tumors are associated with inflammation-related predictors for BRCA (CCDC102B and GLYATL1) and LUSC (SLCO1C1, SLC51B, and BATF2); with invasiveness predictors for LUAC (LY6K and TMEM52); with neurogenesis predictors (LRRC7, LMX1B, ROPN1B) in PRAD; and with hypoxic (FLVCR1) and DNA-damage response (AUNIP) predictors for LIHC. These observations suggest that tumor growth is stimulated by distinct microenvironmental factors in each cancer type.

## Auto-predictors and their properties

Interestingly, we noted that the top predictors do not include genes that have edges leading from a gene in control tissue to the same gene in tumor. We refer to such genes as auto-predictors and asked, if such auto-predictors have properties distinct form the properties of the top predictors. To answer this question, in each cancer, we have identified all auto-predictors (see S5 File for the list of auto-predictors). Note that since all network edges are between differentially expressed genes only, this excludes genes whose expression is simply the same in normal and tumor tissues. Not surprisingly, gene expression in control and tumor samples for auto-predictors tends to be significantly correlated (S5 File). Strikingly, in contrast to the predictors with high predictive potential, GO enrichment analysis of auto-predicting genes revealed relations to oxidation (including hemoglobin complex) and detoxification (Fig 6, S6 File). In particular, the terms enriched for the auto-predictors did not include terms directly related to tissue remodelling. While both, the top predictors and auto-predictors, contained terms related to immune response, the terms for auto-predictive genes were more strongly focused on Major Histocompatibility Complex (MHC)-related immune response (Fig 6, and S6 File). Methods and data used for the functional enrichment are described in Section 'Materials and methods'.

## Discussion and conclusions

The regulation of tumor progression is not fully understood. Recent studies demonstrated that control samples representing adjacent normal tissue in solid tumor studies differ form healthy tissue [11]. We hypothesised that genes whose expression in control samples is predictive of the expression in tumor samples might provide insights into this relation.

We developed and validated a new computational method, TranNet, which is able to identify genes in adjacent normal tissue that are predictive (and potentially facilitating) regulation of a tumor by the tumor environment (Fig 4). In this study we focused on tumor-related genes for both: the predictor and the target gene sets. It is possible that the expression of tumor-unrelated genes in control tissue might be informative about tumor progression. However narrowing down our analysis to these genes allowed us to focus on the most direct tumor driving

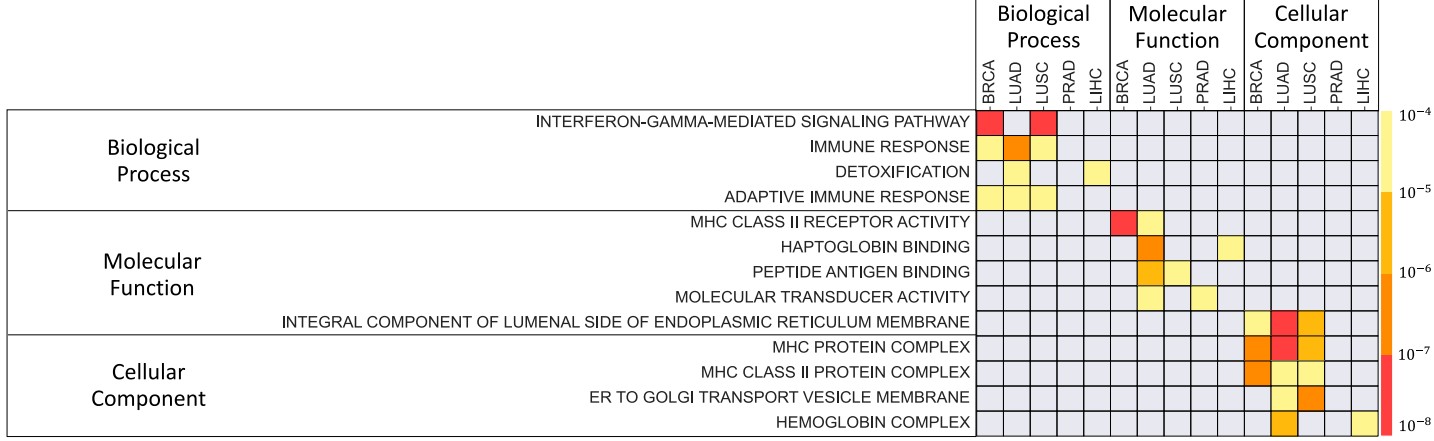

**Fig 6. Functional Enrichment of Auto-Predictors.** GO terms enriched in at least two cancers are shown and the complete lists of GO terms enriched in such auto-predictor genes are provided for the five cancers in S6 File.

relationships. At the same time the variables representing principal components allowed us to capture the remaining relationships. In particular the information about sex has not contributed to the results (Fig F in S1 Text) while nearly all the top marker genes (Table 3) identified by this approach are associated with cancer progression.

We found that the functional properties of the predictor genes are consistent with the mechanisms of regulation by the environment as summarised in Fig 7. In addition, many of the identified predictors are previously recognized markers of the microenvironment-mediated tumor progression. One limitation of this study is that while TranNet identifies predictors of such external influences on a tumor it does not establish causality.

It is important to recognize that some external factors that might impact both normal tissues and tumors and do not necessarily cause tumor progression directly. Even though, many of these factors interact with and/or regulate tumor and are interesting in this respect COPID and hepatitis are likely examples. There is also an association between the incidence of a wide

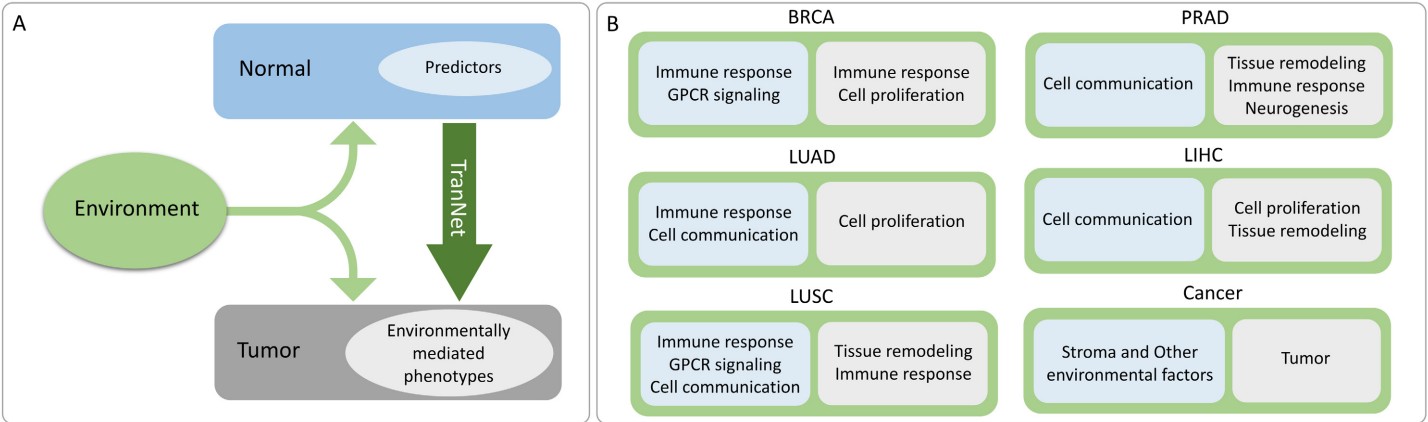

**Fig 7. Interpretations of the relations between tumor and normal tissues inferred by TranNet.** TranNet predictors (light blue) in normal tissue predict environment-related changes in tumor (light grey) **(A)**. Cancer-specific functional enrichment of predictor genes (light blue) and inferred environmentally modulated changes in tumor (light gray) **(B)**.

variety of malignancies and diabetes. Identification of such interactions is of fundamental importance for preventing cancer and development of novel treatments.

The concept of tumor environment, has many layers. Previous studies distinguished tumor's micro and macro environments to capture the effects of the closest neighborhood (such as normal cells, molecules, and blood vessels that surround and feed a tumor cell) and more distant influences respectively [86]. Here we also allow yet another level of influences: organismal environment and germline variations. Our approach provides a proof of principle that control samples can be used to gain insight into the impact of external factors, jointly referred here as the environment, on tumor and tumor progression. The enrichment of the predictor genes in processes associated with immune response suggest that some of these predictors might be in fact sensors of common exposure to chronic infection in normal and tumor tissues. Indeed, it is recognized that cigarette smoking induces lung inflammation, leading to the changes in cellular composition and functions, such as a reduction in the number and properties of ciliated cells. In future studies, it would be interesting to extend the model to allow for a separation of different types of external contributions. In general, it might be difficult to fully deconvolute such processes, as environmental contributors are mostly unknown. However, some environmental factors are mutagenic and in such cases the exposure of the organism to an environmental factor can be estimated by the strength of a corresponding mutational signature [87, 88]. These potential extensions aside, the results obtained by TranNet already provided new insights into the relation between tumor, adjacent control samples, and tumor environment. These relationships might have important implications for the detection and diagnosis of cancers at early stage.

## Materials and methods

### Gene expression data

Batch effect removed TCGA gene expression data [89] were downloaded from GitHub repository (https://github.com/mskcc/RNAseqDB) on April 5th, 2022. Requiring matched normal and cancer samples, five types of EMT cancers: Breast Cancer (BRCA), Lung Adenocarcinoma (LUAD), Lung squamous cell carcinoma (LUSC), Prostate Adenocarcinoma (PRAD), and Liver Hepatocellular Carcinoma (LIHC) were selected for the analysis and only patients that had both normal and tumor gene expressions were kept for each cancer type. We excluded genes which are expressed in neither normal (control) nor tumor samples, and the expression level of a gene is assumed as significant if 80% or more of its expression values over samples are greater than a threshold (see Fig G in S1 Text). We then selected the genes that are differentially expressed between control and tumor samples, and expression data for the remaining genes (non-differentially expressed) are compressed in the principal components that preserve 99% of covariate information of the data [15]. From these principal components (PCs), only differentiated (differentially expressed) components are assumed as additional meta nodes in the network, representing the presence of potential covariates in the transition network, coming from the non-differentially expressed genes [14]. The conceptual idea for defining the differentially expressed principal components and for including them as additional variables in the presenting analysis is described in Section 'Inference of the transition network' and Fig 2B. Finally, activity (expression of genes and magnitudes of PCs) of each node is standardized with Z-score to bring dissimilar features on a similar scale. The data cohorts for all the five cancers are provided as Table 4.

**Table 4. Data summary.** DE genes (tumor genes): differentially expressed genes selected based on T-test q-value < 0.01; Non-DE Genes: genes whose expressions are not differentiated between control and tumor samples; DE PCs: principal components which are differentiated between control and tumor samples representing potential impacts of Non-DE Genes (T-test q-value) on the transition mapping; TranNet: Pairs of Nodes in the Transition Network (DE Genes + DE PCs); Patients: Patients control and tumor samples are available (Clinical cohorts of the patients are summarized in Table B in S1 File); HN Genes: genes in both DE Genes and HumanNet-FN; HN Edges: Edges within HN Genes in the functional network.

| # | DE Genes | Non-DE Genes | DE PCs (q value) | TranNet | Patients | HN Genes | HN Edges |
|---|---|---|---|---|---|---|---|
| BRCA | 9308 | 4831 | PC27(0.0015) | 9309 | 105 | 8633 | 308985 |
| LUAD | 8688 | 5798 | PC3, 6, 8(2.1E-05, 0.0056, 0.0099) | 8691 | 57 | 8087 | 269821 |
| LUSC | 9965 | 4853 | PC5(0.0003) | 9965 | 50 | 9246 | 345514 |
| PRAD | 6398 | 8440 | PC3(4.72E-05) | 6399 | 47 | 5951 | 129963 |
| LIHC | 7514 | 6177 | | 7514 | 40 | 7011 | 230952 |

### Leave-one-out test

The leave-one-out test was performed as follows. For each patient $i$, remove the corresponding expression values $x_{i(\cdot)}$ (a row) and $y_{i(\cdot)}$ (a row) from the data $X$ and $Y$. A transition matrix $M_{(-i)}$ is then computed from the remaining data (such as $X_{(-i)} \times M_{(-i)} \approx Y_{(-i)}$). Then, check if the prediction error on the true response $y_{i(\cdot)}$ is less than the error on the random response $\tilde{y}_{i(\cdot)}$, such that $\|x_{i(\cdot)} \cdot M_{(-i)} - y_{i(\cdot)}\|_{l_1} < \|x_{i(\cdot)} \cdot M_{(-i)} - \tilde{y}_{i(\cdot)}\|_{l_1}$, where $\tilde{y}_{i(\cdot)}$ is obtained as permuting $y_{i(\cdot)}$ ($\|\cdot\|_{l_1}$ is denoted as the absolute summation over the prediction errors on the genes and principal components). A T-test for paired samples was performed to check if this prediction error is less than the random error. The prediction accuracy varies depending on the number of selected predictors as shown in Fig 3 for BRCA, and Fig A in S1 Text for all the five cancers. The number of predictors for each cancer was decided based on different cut-off percents of top genes and principal components with the highest predictive potentials. The test results for the five cancers are summarized in Table 1.

### Functional network

We used a functional network [20] of human genes for disease studies downloaded from HumanNet v3 (https://www.inetbio.org/humannet). This data set contains 18,459 protein coding genes and 977,495 interactions inferred from various datasets and functional networks. The interactions corresponding to our genes sets are used as the edge validation sets in the analysis of the cancers. The characteristics of the gene expression data and edge validation set are summarized in Table 4.

### Functional enrichment in ranked lists of genes

Enrichment of GO term biological processes, molecular functions and cellular components was performed using GOrilla [23] that identifies enriched GO terms in ranked lists of genes. This tool employs a flexible threshold statistical approach to discover GO terms that are significantly enriched at the top of a ranked gene list. The functional enrichment in ranked lists of genes was performed for the target genes sorted by prediction errors (Section 'Insights into the relation between gene expression in tumor and in matched control samples') and the predictor genes sorted by predictive potentials (Section 'Predictor genes are enriched in pathways that imply the features of tumor microenvironment').

### Functional enrichment for subsets of genes

Enrichment of Kegg Pathways (Section 'Pathway enrichment of genes identified as targets of predictors revealed tumor-stroma interactions') and GO terms (Section 'Auto-predictors and

their properties') was performed with a hypergeometric test followed by a Benjamin–Hochberg test for multiple comparison corrections using Kegg pathways [90] and GO terms [91]. All the genes analysed in the study were used as the background, and Kegg pathways and GO terms enriched with $q - value < 0.01$ are reported in this study.

## Supporting information

**S1 Text. Supplemental text.** Tables A and B, Figs A and B and C and D and E and F and G.
(PDF)

**S1 File. Supplemental tables.** Lists of the selected high scoring predictor genes and principal components with their predictive potentials for the five cancers.
(XLSX)

**S2 File. Supplemental tables.** Lists of GO terms enriched for the lists of genes sorted based on prediction errors for each of the five cancers (Genes whose tumor expressions are predicted with smaller errors are listed in higher ranks in the list): Enrichment of biological process, molecular function and cellular component are provided in separate sheets in the same file.
(XLSX)

**S3 File. Supplemental tables.** Lists of GO terms enriched for the lists of genes sorted based on their predictive potential (PP) scores for each of the five types of cancers: Enrichment of biological process, molecular function and cellular component are provided in separate sheets in the same file.
(XLSX)

**S4 File. Supplemental tables.** Lists of enriched Kegg pathways for the target genes of the top 10 PP scoring predictors for the five cancers: Enriched pathways and the corresponding p-values.
(XLSX)

**S5 File. Supplemental tables.** Lists of the auto-predictor genes with their auto-interaction weights, correlation between their normal and tumor tissue expressions and the corresponding p-values.
(XLSX)

**S6 File. Supplemental tables.** Lists of enriched GO terms for the genes whose normal samples are predictive for the corresponding tumor samples (auto-predictors): Enrichment of biological process, molecular function and cellular component.
(XLSX)

## Acknowledgments

We want to thank M.G. Hirsch for comments on the manuscript.

## Author Contributions

**Conceptualization:** Bayarbaatar Amgalan, Teresa M. Przytycka.

**Data curation:** Bayarbaatar Amgalan.

**Formal analysis:** Bayarbaatar Amgalan, Teresa M. Przytycka.

**Investigation:** Bayarbaatar Amgalan, Chi-Ping Day, Teresa M. Przytycka.

**Methodology:** Bayarbaatar Amgalan, Chi-Ping Day, Teresa M. Przytycka.

**Project administration:** Teresa M. Przytycka.

**Resources:** Teresa M. Przytycka.

**Software:** Bayarbaatar Amgalan, Teresa M. Przytycka.

**Validation:** Bayarbaatar Amgalan, Chi-Ping Day, Teresa M. Przytycka.

**Visualization:** Bayarbaatar Amgalan, Chi-Ping Day, Teresa M. Przytycka.

**Writing – original draft:** Bayarbaatar Amgalan, Teresa M. Przytycka.

**Writing – review & editing:** Bayarbaatar Amgalan, Chi-Ping Day, Teresa M. Przytycka.

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
