## [Decision Letter · Decision Letter 0]

23 Aug 2023

Dear Dr. Przytycka,

We are pleased to inform you that your manuscript 'Exploring tumor-normal cross-talk with TranNet: role of the environment in tumor progression' has been provisionally accepted for publication in PLOS Computational Biology.

Best regards,

Giulio Caravagna

Guest Editor

PLOS Computational Biology

Feilim Mac Gabhann

Editor-in-Chief

PLOS Computational Biology

Reviewer's Responses to Questions

**Comments to the Authors:**

Reviewer #1: I thank the reviewers for addressing all my comments.

I have no further concern.

Reviewer #2: The authors addressed all the comments by the reviewers, and provided clear responses in this version of the manuscript. I have no further requests.

**Have the authors made all data and (if applicable) computational code underlying the findings in their manuscript fully available?**

Reviewer #1: None

Reviewer #2: Yes

PLOS authors have the option to publish the peer review history of their article (what does this mean?). If published, this will include your full peer review and any attached files.

Reviewer #1: No

Reviewer #2: No

---

## [Editor Report · Acceptance letter]

13 Sep 2023

PCOMPBIOL-D-23-01089 

Exploring tumor-normal cross-talk with TranNet: role of the environment in tumor progression

Dear Dr Przytycka,

I am pleased to inform you that your manuscript has been formally accepted for publication in PLOS Computational Biology. Your manuscript is now with our production department and you will be notified of the publication date in due course.

With kind regards,

Zsofia Freund
